# Nicotinamide Counteracts the Detrimental Effect of Endothelin-1 on Uterine Decidualization During Early Pregnancy by Influencing EDNRB

**DOI:** 10.3390/cells14211645

**Published:** 2025-10-22

**Authors:** Yuye Wang, Qing Ma, Meitong Chen, Yukako Kayashima, Jiayi Zhou, Balaji Rao, Jessica L. Bowser, Xianwen Yi, Nobuyo Maeda-Smithies, Feng Li

**Affiliations:** 1Department of Pathology and Laboratory Medicine, The University of North Carolina, Chapel Hill, NC 27599, USA; yuye_wang@med.unc.edu (Y.W.); maqing@email.unc.edu (Q.M.); cmeitong@email.unc.edu (M.C.); yukaya@email.unc.edu (Y.K.); jlbowser@email.unc.edu (J.L.B.); xianwen_yi@med.unc.edu (X.Y.); nobuyo@med.unc.edu (N.M.-S.); 2Department of Nutrition, Gillings School of Global Public Health, University of North Carolina at Chapel Hill, Chapel Hill, NC 27599, USA; jz0105@email.unc.edu; 3Department of Chemical and Biomolecular Engineering, Golden LEAF Biomanufacturing Training and Education Center, North Carolina State University, Raleigh, NC 27695, USA; bmrao@ncsu.edu

**Keywords:** Endothelin-1, decidualization, nicotinamide, mice, preeclampsia, human endometrial stomal cells

## Abstract

**Highlights:**

**What are the main findings?**

**What is the implication of the main finding?**

**Abstract:**

Endothelin-1 (ET-1) is involved in the pathogenesis of preeclampsia. Mice (*Edn1^H/+^)* having excess endothelin-1 developed preeclampsia-like phenotypes during pregnancy in a maternal genotype-dependent manner. Here, we investigated whether decidualization is impaired in *Edn1^H/+^* dams, and whether nicotinamide (a potent inhibitor of ET-1) exerts beneficial effect. We compared implantation sites between wild type (WT) and *Edn1^H/+^* dams with or without nicotinamide treatment. Implantation sites of *Edn1^H/+^* dams exhibited abnormal ectoplacental cones and sinusoids, along with reduced vascular density in the mesometrial regions of the decidua. VEGF levels were higher in the decidua of *Edn1^H/+^* dams compared with WT dams. Markers of decidualization were decreased in *Edn1^H/+^* dams. Nicotinamide supplementation corrected this abnormality. During differentiation (decidualization) of cultured human endometrial stomal cells, ET-1 impaired the upregulated expression of decidualization markers. The effect of ET-1 was reversed by nicotinamide. These results show nicotinamide counteracts the detrimental effects of ET-1 on endometrial decidualization and has potential to improve embryo implantation and subsequent pregnancy outcomes.

## 1. Background

Endothelin-1 (ET-1), a 21-amino acid peptide, is one of the most potent vasoconstrictors identified. ET-1 exerts its biological functions through two receptors, EDNRA and EDNRB, and can have distinct effects on different cell types [1].

ET-1 plays an important role in the pathogenesis of pregnancy complications including preeclampsia (PE) [2], characterized by hypertension, proteinuria and end-organ damage [3]. There is an association between *EDN1* (gene encoding pre-pro-ET-1) G5665T (Lys198Asn, rs5370) polymorphism and PE [4,5]. In animal studies, Greenberg et al. administered ET-1 to pregnant sheep through continuous intravenous infusion and found that elevated ET-1 plasma levels over a 4 h period resulted in the onset of hypertension, renal and uterine vasoconstriction, and proteinuria mimicking the phenotype of PE [6]. Our prior study reported that female *Edn1^H/+^* mice, having ~2–3× higher plasma ET-1 levels than wild-type (WT) mice, developed the full-spectrum PE-like phenotypes during the third week of pregnancy in a maternal genotype-dependent manner [7]. In addition, we found that embryo implantation was impaired in *Edn1^H/+^* dams. For example, all the embryo implantation sites regardless of their genotype/sex had a disoriented ectoplacental cone (EPC) and sustained elevated expression of E-cadherin (evidence of poor trophoblast cell invasion) at 7.5 days post coitus (dpc) [7]. These findings indicate that the impaired implantation results from high maternal *Edn1* expression, independent of embryo genotype. However, the precise role of ET-1 in uterine decidualization, which plays a pivotal role in implantation, is largely unknown.

Nicotinamide (amide form of vitamin B3, NAM) is a potent inhibitor of ET-1 downstream of ADP ribosylcyclase [8,9]. We tested this agent in two different mouse models of PE [one with experimentally induced sFLT1 overexpression and another genetically lacking ASB4 (Ankiryn-repeat-and-SOCS-box-containing-protein [10])], and reported that NAM treatment throughout the entire pregnancy decreased blood pressure (BP) and renal injury and prolongs pregnancy [11]. NAM ameliorated PE-like conditions in dams with reduced uterine perfusion pressure (RUPP) as well [12]. Thus, NAM has the potential to reverse serious maternal and fetal symptoms of PE.

Impaired implantation leading to insufficient spiral artery remodeling plays a central role in the pathogenesis of PE. However, whether and how NAM would prevent the problems during the early implantation stage of pregnancy has not been investigated.

In the current study, we investigated the role of ET-1/NAM in endometrial decidualization during the early stages of pregnancy, when implantation occurs, using *Edn1^H/+^* mice and cultured human endometrial stromal cells. Our results demonstrate that NAM mitigates the detrimental effects of maternal ET-1 on endometrial decidualization during early pregnancy.

## 2. Materials and Methods

### 2.1. Mice

Mice (C57BL/6J) including both sexes and genotypes (WT and *Edn1^H/+^*) were housed in standard cages on a 12 h light/dark cycle and were allowed free access to food and water. All experiments were carried out in accordance with the National Institutes of Health guideline for use and care of experimental animals, as approved by the IACUC of the University of North Carolina at Chapel Hill (protocol #24-181).

### 2.2. Matings

At the age of 8–10 weeks, litter mates of wild-type (WT) and *Edn1^H/+^* female mice were mated with WT males. Females were checked for vaginal plugs each morning, and the day of plug detection was designated as 0.5 days post coitus (dpc).

### 2.3. Three-Dimensional (3D) High-Frequency Ultrasonography (HFU)

At 7.5 dpc, dams were anesthetized with 2% isoflurane (Isospire™, Dechra Veterinary Products, Overland Park, KS, USA). Abdominal hair was removed with depilatory cream (Nair^TM^, Church & Dwight Co., Trenton, NJ, USA). Implantation site detection and location within each uterine horn were visualized transabdominally using the VisualSonics VevoF2 Imaging System with a 57× transducer (FUJIFILM VisualSonics Inc., Toronto, ON, Canada) at 55 megahertz. The 3D mode was used for advanced data acquisition and analysis, with virtual sections obtained in all directions (x-, y-, z-). The scan distance was set at 10.2 mm, with a step size of 0.08 mm and a total of 127–133 frames were captured per 3D scan. Each ultrasound examination was finished within ~15 min; heart rate and body temperature were monitored during the procedure [13,14].

### 2.4. Morphological Examination

The implantation sites of 7.5 dpc were dissected and examined visually, and the number of sites was counted. Fixed implantation site tissues (4% paraformaldehyde) were sectioned (5 μm) and stained with hematoxylin and eosin (H&E) [7].

### 2.5. Decidua Preparation

One implantation site (7.5 dpc) from each dam of four groups (WT and *Edn1^H/+^* dams with or without NAM) was randomly collected. The embryo and ectoplacental cone were carefully removed and the remaining maternal decidua (illustrated in Appendix A) was subjected to either RAN isolation as described below (Quantitative RT-PCR section) or homogenization in 0.5 mL buffer (0.1% Triton in PBS) for future Western blot/ELISA assay [15].

### 2.6. Biochemical Analyses

The ELISA kit for measuring ET-1, VEGF, and BMP2 was purchased from R&D Systems, Inc. (DET100, MMV00, DBP200, respectively). The ELISA kit for estrogen was from abcam (#ab285291, Fremont, CA, USA), and progesterone was from CrystalChem (#80559, Elk, Grove Village, IL, USA).

### 2.7. Western Blot

Total protein (20 µg/lane) were subjected to 4–20% SDS-PAGE, then electrotransferred onto PVDF membranes. The chemiluminescent intensities of the targeted protein bands were captured using the ODYSSEY^®^ FC system and evaluated using Image Studio Software 2.0 (LI-COR Biosciences, Lincoln, NE, USA). The individual protein level of each sample was quantified by normalizing its intensity to the β-actin in the same sample and expressed relative to the levels of the respective control (WT without NAM) group, the mean of which was set as 1 [16]. The antibodies used in the study included prolactin (#PA5-119669, ThermoFisher, Waltham, MA, USA) and β-actin (#5125, Cell Signaling Technology Inc., Danvers, MA, USA). The primary antibody was diluted 1:500 and incubated overnight at 4 °C and secondary antibody was diluted 1:1000 and incubated for 1-h at room temperature.

### 2.8. Immunohistochemistry (IHC)

The antibodies used were CD31 (#AF3628, Cell Signaling Technology Inc. Danvers, MA, USA), VEGF (#CME356AK, BioCare Medical, Pacheco, CA, USA), BMP2 (#MA5-29055, ThermoFisher, Waltham, MA, USA), cytokeratin 17 (#12509, Cell Signaling Technology Inc., Danvers, MA, USA). IHC was performed in a Bond fully automated slide staining system (Leica Bicrosystems Inc., Vista, CA, USA). Antigen retrieval was performed at 100 °C in Bond-epitope retrieval solution 1 pH 6.0 (AR9961) for 20 min. Positive and negative controls (no primary antibody) were included for each antibody. IHC stained sections were digitally imaged (20× objective) in the Aperio ScanScope XT using line-scan camera technology (Leica Biosystems, Vista, CA, USA) [7].

### 2.9. Cell Culture

Immortalized human endometrial stromal cells (HESC) were obtained from Applied Biological Materials Inc. (T0533, Vancouver, BC, Canada) and maintained following recommended instructions by the company. For differentiation, cells were treated with a decidualization cocktail (DC) containing 1 μM medroxyprogesterone acetate (#461120050, ThermoFisher, Waltham, MA, USA), 10 nM estradiol (#436320050, ThermoFisher, Waltham, MA, USA), and 0.5 mM 8-bromo-cAMP (#14431, Cayman, Ann Arbor, MI, USA) [17,18] in PriGrowIV media without phenol red (#TM004, Applied Biological Materials Inc., Vancouver, BC, Canada) supplemented with 2% charcoal-stripped calf serum (#A3382102, ThermoFisher, Waltham, MA, USA) and varying concentrations of ET-1 (0, 0.01, 0.1, 1 μM, #1160, Tocris Bioscience, Bristol, UK) for 3 days. Then bulk RNA-sequencing was performed. After determining the optimal dose of ET-1, HESC’s were treated with DC, ET-1 (0.01 µM) and ET-1 plus varying doses of NAM (0.1, 1, and 10 mM) for 3 days. Then qRT-PCR and bulk RNA-sequencing were performed.

Different batches of HESC cells were treated with DC, DC + ET-1 (0.01 µM), DC + ET-1 + NAM (10 mM) or DC + ET-1 + NAM (10 mM) + BQ788 (1 μM [7]) for 6 days (medium was changed every other day), then, morphological changes were examined by light microscope (Nikon, Digital Sight 1000, Melville, NY, USA). Experiments were repeated three times with each experimental group consisting of 3–4 replicates.

HESC cells were treated with NAM (10 mM) without DC for 6 days, and morphological changes and gene expression were determined. Experiments were repeated three times with each experimental group consisting of three replicates.

### 2.10. Assessment of Motility of HESCs with Wound Healing Assay

When HESCs reached 100% confluence in a 24-well plate (indicated as day 0), cells were scratched along the midline of the well with a 200 µL pipette tip (#4845, Universal Fit Pipet Tips, Corning, NY, USA, approximately 600 µm in diameter) and images were captured using a Nikon Digital Sight 1000 camera [19]. Scratched cells were then treated with the different medium as described above. After 72 h of exposure to different treatments, cell migration into the wound was visualized using the same microscope again [20,21]. Data was presented as Δ wound width (wound width at day 0 − wound width at day 3). Experiments were repeated three times with each experimental group consisting of 3–4 replicates.

### 2.11. Trophoblast Spheroid Formation

HTR-8/SVneo cells [7] were cultured in RPMI 1640 medium (Corning, Manassas, VA, USA) supplemented with 1% Penn/Strep and 5% FBS until 80% confluence. HTR-8 cells were then trypsinized and diluted to 300 cells/µL in RPMI 1640 medium with 5% FBS. Cell counting was performed using a hemacytometer (Hausser Scientific, Horsham, PA, USA). Approximately 6000 HTR-8 cells in 20 µL were transferred to an ultra-low cluster 96-well plate with round bottom (#7007, Costar, Kennebunk, ME, USA) and incubated at 37 °C degrees for 16–18 h [22].

### 2.12. Spheroid Attachment to Epithelial Cells

Ishikawa cells [23] were plated in 24-well plates with MEM (#11095080 ThermoFisher, Waltham, MA, USA) with 10% FBS, 1 mM sodium pyruvate, 0.1 mM non-essential amino acids, 1% Penn/Strep until reaching 100% confluence. Cells were starved for 24 h with 0% FBS medium. The respective treatment groups are: 1. Control (0% FBS medium), 2. 0.5 µM ET-1 in 0% FBS media. After 24 h of treatment, 10 trophoblast spheroids were transferred into each well. After 24 h of incubation, media were removed and replaced with PBS (Corning, Manassas, VA, USA). Plates were vigorously tapped and shaken to remove the unattached spheroids. The attached spheroids were counted under microscope [24]. Data was presented as percentage of attached spheroids: (the number of remaining spheroids/10) × 100.

### 2.13. RNA Sequencing

Total RNA from different groups of HESCs was extracted using RNeasy mini kit (Qiagen, Hilden, Germany) according to the manufacturer’s protocol. After the quality of RNA was examined, the High-Throughput Sequencing Facility (HTSF) at UNC-CH performed library preparation and sequencing. Paired-end raw sequences of 100 bp were acquired by NextSeq 2000 (Illumina, San Diego, CA, USA), and subsequent demultiplexing was performed using bclconvert with a 1-mismatch threshold to segregate reads based on indexes. Raw sequencing reads were preprocessed using FASTP to remove adapters and trim poly-G tails. The trimmed reads were then aligned to the Ensemble gene database (release 113) using the SALMON aligner. The gene count matrices were generated and extracted from the SALMON quantification files. The raw counts were then filtered and normalized for the subsequent differential gene expression analysis using relevant Bioconductor packages in R studios (Posit Software, version 4.5.1). The data was deposited to GEO (GSE306844).

### 2.14. Quantitative RT-PCR

Total RNA from tissues or cells was extracted using Trizol (Life Technologies, St. Paul, MN, USA) following the manufacturer’s instruction. A NanoDrop spectrophotometer method and gel electrophoresis were used to check the quantity and quality of RNA. mRNA was quantified with QuantStudio 3 Real-Time PCR system (Thermo Fisher, Waltham, MA, USA) by using one-step RT-PCR Kit (Bio Rad, Hercules, CA, USA) with *18s* as reference genes in each reaction for mouse tissue. For human cell experiments, *GAPDH* was used as a reference gene [7,25]. The primer and probe sequences are in Appendix A.

### 2.15. Statistical Analysis

Data are presented as mean ± SEM. A multifactorial ANOVA test was used with the program JMP 17.0 (SAS Institute Inc., Cary, NC, USA). Post hoc analyses were done using the Tukey–Kramer Honest Significant Difference test or Student *t*-test as described in figure legends. Two-tailed analysis is applied unless otherwise indicated. Differences were considered to be statistically significant with *p* values less than 0.05.

## 3. Results

### 3.1. The Decrease in Implantation Site Volume in Edn1^H/+^ Dams Is Prevented by NAM Treatment

In order to compare the implantation site volume between WT and *Edn1^H/+^* dams, 3D HFU analysis was used and revealed that the implantation site volume at 7.5 dpc from *Edn1^H/+^* dams was ~0.5× lower than that from WT dams. NAM treatment increased the implantation site volume by ~1.5× in *Edn1^H/+^* dams but did not affect WT dams (Figure 1A,B).

Visually examining the implantation sites from the four groups of dams showed that *Edn1^H/+^* dams had significantly fewer implantation sites than WT dams (average 5.3 in *Edn1^H/+^* dams vs. 7.8 in WT dams, *p* = 0.01) (Figure 1C). NAM tended to increase about one implantation site per *Edn1^H/+^* dam but did not reach significant levels (*p* = 0.06). NAM had no effect on WT dams (Figure 1D). These results suggest that NAM improves embryo development in *Edn1^H/+^* dams.

### 3.2. The Impaired Structure of Implantation Sites from Edn1^H/+^ Dams Is Improved by NAM

H&E staining and CD31 (PECAM-1, marker of endothelial cells) immunostaining of implantation sites from *Edn1^H/+^* dams at 7.5 dpc showed a reduced vascular sinus folding (VSF) region in the mesometrial region (MR) compared with those from WT dams (Appendix A illustrates the structure of an implantation site). NAM treatment improved the altered decidual vascular structure in *Edn1^H/+^* dams but had no effects on WT dams (Figure 2A,B). Semi-quantification analysis showed decreased CD31^+^ area in *Edn1^H/+^* dams compared with WT dams. NAM increased the area in *Edn1^H/+^* dams but did not affect WT dams (Figure 2C).

Embryos from *Edn1^H/+^* dams had distorted ectoplacental cones (EPCs), as we previously reported [7], and NAM corrected its orientation (Figure 2D).

### 3.3. NAM Treatment Improves the Structure of Maternal–Embryo Interface During Early Pregnancy

The increased expression of vascular endothelial growth factor-A (VEGF) in the sinusoids of the mesometrial region of *Edn1^H/+^* dams is mitigated by NAM.

Excess VEGF alters the decidual vascular network [26] and our previous work showed that *Edn1^H/+^* mice had higher plasma VEGF levels than WT mice [7]. Therefore, we determined the expression of VEGF in the deciduae (Appendix A illustrates the decidual region used for experiments henceforth). Immunostaining and ELISA showed that VEGF were higher in the deciduae of *Edn1^H/+^* dams compared with WT dams. NAM treatment decreased VEGF in *Edn1^H/+^* dams but did not affect WT dams (Figure 3A,B). The mRNA levels of *Vegf* also had the same pattern among the four groups of dams (Figure 3C). However, NAM treatment did not significantly decrease plasma VEGF concentration. (Figure 3D). These results suggest that the effect of NAM treatment on VEGF is likely localized to the deciduae.

There is a report that ET-1 increases VEGF through Hypoxia-inducible factor 1-alpha (HIF-1α) [27]. HIF-1α mRNA levels in deciduae were measured, and the pattern of its expression was in parallel with that of VEGF in deciduae from four groups of mice. (Figure 3E).

### 3.4. Decidualization Is Impaired in Edn1^H/+^ Dams, Which Is Mitigated by NAM

Decidualized endometrial stromal cells contribute to angiogenesis, which is essential for decidualization [28,29]. Accordingly, we investigated the effects of ET-1 on uterine decidualization. Prolactin expression (PRL, a marker of endometrial decidualization), determined by western blot, was reduced in the deciduae of *Edn1^H/+^* dams than WT dams. NAM treatment increased PRL expression in *Edn1^H/+^* while NAM had no effect on PRL levels in WT dams (Figure 4A,B).

We next performed immunoassays for BMP2 (bone morphogenetic protein 2, a marker of endometrial decidualization). We found BMP2 levels were less in the deciduae of *Edn1^H/+^* dams compared with WT dams. NAM treatment increased BMP2 in *Edn1^H/+^* while NAM did not affect BMP2 in WT dams (Figure 4C).

There is an interplay between ET-1 and progesterone [30,31,32], and progesterone is reported to play a pivotal role in endometrial decidualization [33,34]. We measured plasma levels of progesterone and found there was no difference among the four groups of mice (Appendix A). We measured plasma levels of estrogen as well because it plays a role in stromal cell differentiation [35]. The plasma levels of estrogen tended to be higher in *Edn1^H/+^* dams compared with WT dams. NAM treatment increased estrogen plasma levels in WT dams yet had no effect on *Edn1^H/+^* dams (Appendix A).

These results suggest that ET-1 directly impacts endometrial decidualization in the decidua.

### 3.5. NAM Inhibits ET-1 Expression in the Deciduae of Edn1^H/+^ Dams

Because ET-1 directly regulates its own secretion [36,37] and NAM blocks ET-1 signaling, NAM could alter ET-1 and its two receptors. We examined the expression of ET-1, and two receptors in the deciduae. First, as expected, *Edn1* (mouse gene encoding precursor of ET-1) mRNA levels were markedly higher in control *Edn1^H/+^* dams than their WT counterparts. NAM decreased *Edn1* expression in the deciduae from *Edn1^H/+^* dams (Figure 5A). Furthermore, ET-1 protein levels in the deciduae and plasma from the four groups of dams were measured by ELISA. *Edn1^H/+^* dams had elevated ET-1 levels in the deciduae and circulation, both of which were reduced by NAM (Figure 5B,C). In contrast, NAM did not change ET-1 levels in WT dams. Lastly, we found the mRNA levels of *Ednra* (encoding ET type A receptor) and *Ednrb* (encoding ET type B receptor) in the deciduae were also higher in control *Edn1^H/+^* dams than their WT counterparts. NAM decreased the expression of *Ednrb* but not *Ednra* in *Edn1^H/+^* dams (Figure 5D,E). NAM did not affect the mRNA levels of *Ednra* and *Ednrb* in WT dams (Figure 5D,E).

### 3.6. ET-1 Inhibits the Endometrial Stromal Cell Differentiation In Vitro, Which Was Corrected by NAM

Endometrial stromal cell decidualization (differentiation) is an essential prerequisite for implantation, refs. [33,34] and stromal cells are the most abundant cell types in the deciduae [38]. Therefore, we tested the effects of ET-1 at three concentrations (0.01, 0.1 and 1 µM) on human endometrial stromal cell differentiation in vitro. We found that all three doses of ET-1 suppressed the upregulation of differentiation markers (WNT4, IGFBP-1) and promoted the downregulation of proliferation markers (Ki67, ID2) three days after being treated with a cell differentiation cocktail (Appendix A). Following these results, we used 0.01 µM of ET-1 in the following experiments. To investigate whether NAM could counteract the inhibitory effects of ET-1 on differentiation, we tested 0.1(L), 1(M) and 10 (H) mM of NAM and found that NAM abolished ET-1 effects in a dose dependent manner, 10 mM of NAM dramatically increased the mRNA levels of markers of decidualization (Figure 6A–C). This was consistent with RNA-seq analysis which revealed the upregulated expression of genes during decidualization was decreased by ET-1; however, NAM restored the expression of these genes (Appendix A).

We next examined the effects of ET-1/NAM on stromal cell morphological change during decidualization. Six days after differentiation, the cells in DC groups gained the morphological features of decidual cells (e.g., larger and multinucleated) [39,40] (Figure 6D and Appendix A), while the cells in DC+ET-1 group kept the typical fibroblastic characteristics similar to non-differentiated control cells. NAM improved the cell differentiation in DC+ET-1 group as evidenced by the presence of larger and multinucleated cells (Figure 6D and Appendix A).

Functional effects of ET-1/MAN on stromal cells were evaluated by cell motility (wound healing) experiments. Stromal cells exposed to DC had increased cell motility, which was inhibited by ET-1 but corrected by NAM (Figure 6E,F).

To determine if NAM has effects on non-decidualized stromal cells, cells were treated with NAM at dose of 10 mM. At this high dose, NAM did not alter the mRNA levels of PRL and WNT4 (Appendix A) and did not alter the morphology of the cells (Appendix A). NAM only increased the mRNA levels of IGFBP1 slightly compared with its effects on decidualized stomal cells (Appendix A).

### 3.7. The Beneficial Effects of NAM on Stromal Cell Decidualization Are Mediated Through EDNRB

RNA-seq analysis showed that the expression of *EDN1* and *EDNRA* was detected but *EDNRB*’s expression was nondetectable in non-differentiated stromal cells (Appendix A). However, the three major molecules in ET-1 signaling were upregulated three days after differentiation (Appendix A). Exposure to ET-1 further increased *EDN1* expression, but NAM decreased *EDN1* expression in a dose-dependent manner. While neither ET-1 nor NAM had any effects on *EDNRA*, ET-1 tended to decrease *EDNRB* expression and NAM corrected its expression in a dose-dependent manner.

Following this, we tested the effect of selective EDNRB antagonist (BQ788) on the beneficial role of NAM in decidualization. Addition of BQ788 to medium containing DC, ET-1, and BQ788 abolished NAM’s promoting role in decidualization (Figure 7A), as evidenced by reduced expression of decidualization markers, impaired cell motility (Figure 7B,C) and loss of decidualized stromal cell morphology (Figure 7D).

### 3.8. Excess ET-1 Does Not Alter the Receptivity of Uterine Luminal Epithelial Cells (Ishikawa Cells)

Endometrial epithelial cells provide the initial receptive surface for attachment. If this process is disturbed, embryo implantation is also impaired. Ishikawa cells are widely used in vitro models to study embryo implantation [41]. We found that Ishikawa cells treated with 0.5 µM of ET-1 did not decrease the attachment of trophoblast spheroids compared to non-treated control cells (Appendix A).

## 4. Discussion

In the current study, we have demonstrated that the increase in ET-1 expression causes impaired endometrial decidualization and angiogenesis at the early stage of pregnancy when implantation/invasion occurs. NAM treatment mitigated these detrimental effects observed in dams having excess ET-1. Our in vitro data showed that ET-1 inhibits endometrial stromal cell differentiation, as evidenced by (1) the upregulating expression of decidualization markers being blocked by the addition of ET-1 in differentiation medium, and (2) exposure to ET-1 causing poor morphological change and motility of stromal cells. NAM counteracted the effect of ET-1 in a dose-dependent manner in this cell culture model. Our results suggest that ET-1 interferes with endometrial decidualization, while NAM (an inhibitor of ET-1 signaling) mitigates these detrimental effects of ET-1.

Embryo implantation is a highly orchestrated process which requires appropriate maternal-embryo interaction. Previously, we found that the embryos from *Edn1^H/+^* dams, regardless of their genotypes, were developmentally delayed about 12 h and exhibited distortion of ectoplacental cones (EPCs) at 7.5 dpc, the initial stage of trophoblast invasion [7]. Thus, it is highly possible that high expression of ET-1 in the maternal tissues leads to decidualization problems which affect embryo implantation. Therefore, we focused on the endometrial decidualization in the current study. The ability of endometrial stromal cells to decidualize along with the development of the vascular network is critical for successful pregnancy. Prior studies demonstrating the stromal cells express ET-1 and its receptors [42,43], along with our RNA-seq data showing that the upregulation of *EDN1*, *EDNRA* and *EDNRB* after 3 days of differentiation suggest that ET-1 signaling plays a fundamental role in decidualization. Here, we found that mice with high ET-1 had impaired decidualization during early stage of pregnancy and addition of ET-1 in the differentiation medium caused insufficient differentiation of stromal cells, which were corrected by NAM. These data provide evidence that the ET-1 system directly impacts on decidualization under physiological/pathological conditions.

Strikingly, excess ET-1 downregulated *EDNRB* while it increased its own expression but had no effect on *EDNRA* during stromal cell differentiation. In contrast, NAM abolished the effects of ET-1 on *EDNRB* and itself in a dose-dependent manner without altering *EDNRA*. The data suggests that EDNRB signaling plays an important role in stromal cell differentiation. Indeed, BQ788 (a selective EDNRB antagonist) blocked the counteracting effects of NAM on ET-1. Activation of EDNRB triggers signaling pathways including PI3K, calcium signaling, eNOS/NO, MAPK/src kinase/PKC, among others. Identifying which factor(s) mediates the effects of NAM/EDNRB against ET-1 overactivation on stromal cell differentiation warrants future investigations.

scRNA-seq analysis of implantation site of WT mouse uterus at the invasion phase of embryo implantation approximately 7.5 dpc revealed multiple types of cells including stromal cell, epithelial cells, endothelial cells, immune cells [44]. All these types of cells express EDNRB, however, the relative expression of EDNRB in normal or preeclamptic pregnancy is not clear. The in vitro data we obtained here suggests that EDNRB expression was upregulated during normal stromal cell decidualization and this upregulation was inhibited by ET-1 exposure. In contrast, in vivo data showed *Edn1^H/+^* dams had increased decidual EDNRB expression. Taken together, these data suggest EDNRB signaling plays distinct roles in different types of cells in normal pregnancy or PE, especially in a subset of PE associated with abnormal high ET-1.

Furthermore, NAM treatment decreased EDNRB in the deciduae of *Edn1^H/+^* dams, but NAM increased EDNRB expression in the cultured endometrial stromal cells during decidualization. Collectively, our results support the notion that ET-1 executes different effects on different types of cells through its two receptors. Due to this, NAM’s effects on ET-1 and two receptors vary across different cell types. Nevertheless, the overall effect of NAM is beneficial and possibly through modulating EDNRB-related signaling pathways. Comprehensive stereological and/or single cell RNA-seq assessment of the effects of ET-1/NAM on different compartments of the deciduae will elucidate how ET-1/NAM influence each type of cells and their crosstalk, which will provide insight into the mechanisms underlying NAM’s beneficial effects.

Our data are consistent with Charron et al.’s discovery which showed ET-1 increases its own expression [45]. Activation of RhoA and p38 MAPK by ET-1 leads to the phosphorylation and activation of transcription GATA-4 which bind −135 bp in the *EDN1* promoter resulting in stimulating *EDN1* expression [46]. Both EDNRA and EDNRB are thought to be involved in this event [47,48]. Our data show NAM downregulated *EDN1* and *EDNRB* expression parallelly but has no effect on *EDNRA,* indicating that *EDNRB* may be the main receptor regulating the ET-1 expression in our experimental setting.

It was reported that ET-1 increases VEGF through HIF-1α [49]. Consistent with this, we found that maternal deciduae from *Edn1^H/+^* dams had higher HIF-1α and VEGF levels than those from WT dams. NAM decreased HIF-1α and VEGF in the deciduae from *Edn1^H/+^* dams parallelly. VEGF is tightly regulated under physiological conditions, and both abnormally high and low VEGF cause poor angiogenesis [26,50]. NAM could decrease VEGF through blocking ET-1-HIF-1α signaling to normalize angiogenesis. We were tempted to test the effects of ET-1 and/or NAM on VEGF using cultured human and mouse endothelial cell lines which are EA.hy926, a human umbilical vein endothelial cell (HUVEC) line originally immortalized by fusion with an A549 human lung epithelial line [51] and bEnd.3 isolated from brain tissue derived from a mouse with endothelioma [25]. Unfortunately, we did not observe any effects of ET-1 and NAM on VEGF. It is highly possible that cell lines we used cannot faithfully represent the features of decidual endothelial cells. In the future, establishing primary cultured decidual endothelial cells could provide a platform to test our hypothesis. It is also possible that other types of cells such as macrophages which produce VEGF [52] are affected by ET-1/NAM leading to the alteration of VEGF we observed in this study.

Both endometrial epithelial cells and stromal cells play an important role in embryo implantation. [53,54] Endometrial epithelial cells provide the initial receptive surface for attachment, while endometrial stomal cells undergo changes (decidualization) to facilitate the blastocyst’s deeper penetration/invasion and ensure a successful implantation [55]. Our data obtained from trophoblast spheroid attachment experiments, showing attachment to epithelial cells (Ishikawa) [24] treated with ET-1, was not altered. Taken together, it is likely that ET-1 affects endometrial stromal cells more than luminal epithelial cells.

In the current study, NAM had no effect on WT dams which was consistent with our prior reports [11,56] and others [12,57]. The known mechanism by which NAM exerts its beneficial effects is suppressing the mobilization of Ca^2+^ from sarcoplasmic reticulum into cytoplasm through inhibiting ADP-ribose cyclase activated by ET-1 [8,9]. The potential mechanism we proposed in this study is that NAM corrects EDNRB expression. In WT dams, ET-1 and its downstream ADP-ribose cyclase and EDNRB expression are in baseline levels (not activated), so there is no elevated mechanism for NAM to counteract. This is a possible reason that there is no obvious effect of NAM on WT dams, as there is no underlying issue for NAM to correct.

In summary, our current study demonstrated that ET-1 inhibits endometrial decidualization during the early stage of pregnancy when embryo implantation/invasion occurs. NAM treatment improves decidualization and angiogenesis through altering ET-1/EDNRB signaling. NAM supplementation has potential to improve decidualization and subsequent embryo implantation, which could lead to improved pregnancy outcomes, especially in pregnancy complications associated with implantation problems including PE.

## Figures and Tables

**Figure 1 cells-14-01645-f001:**
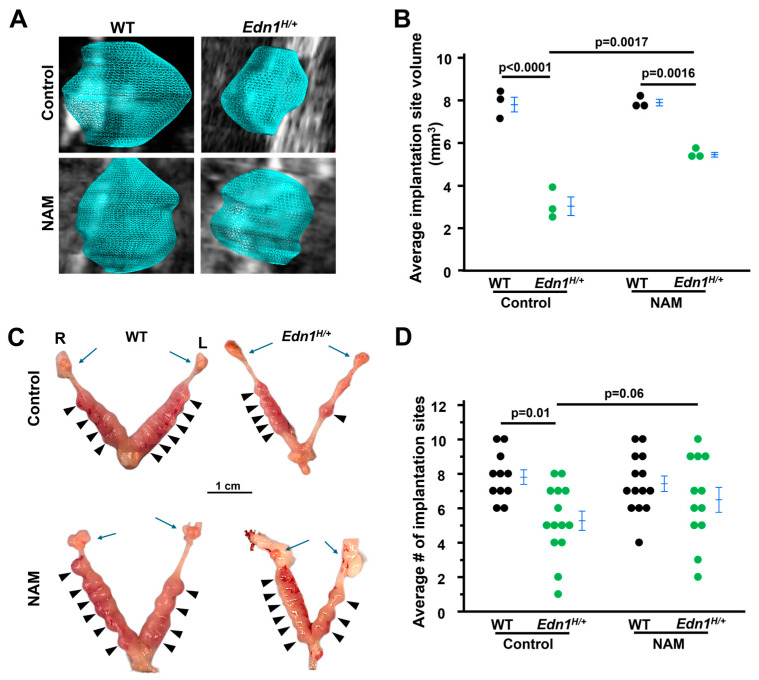
Nicotinamide (NAM) increases the size of implantation sites from *Edn1^H/+^* dams at 7.5 dpc. (**A**) Representative 3D visual reconstruction of implantation sites from four groups mice. (**B**) Average implantation site volume from four groups of dams. 3D volume reconstruction calculations demonstrated increased the volume of implantation sites of *Edn1^H/+^* dams. Each point indicated the average volume of all the implantation sites detected in an individual dam. *n* = 3. Tukey–Kramer HSD. (**C**) NAM increased the size of implantation sites of *Edn1^H/+^* dams. Images of embryos within uteri. R: right uterine horn, L: left uterine horn. Blue arrows: ovaries. Arrow head: viable implantation sites. (**D**) The average number of implantation sites. NAM tended to increase the implantation site number in *Edn1^H/+^* dams: *p* = 0.06, *Edn1^H/+^* vs. *Edn1^H/+^* + NAM, one-tailed *t*-test. *n* = 11–14 dams/group.

**Figure 2 cells-14-01645-f002:**
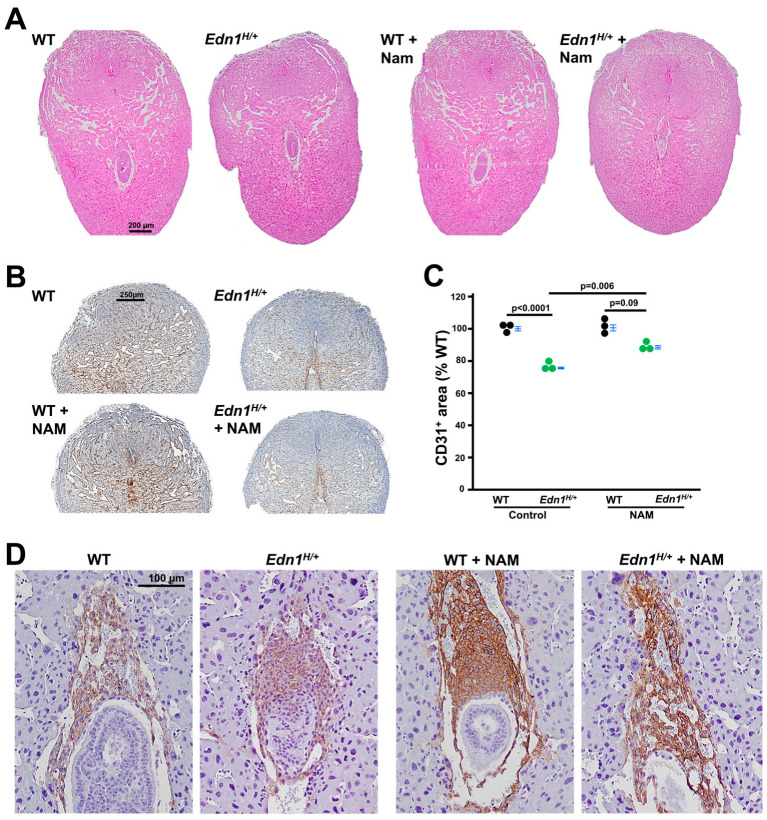
Nicotinamide (NAM) improves the altered morphology of implantation sites from *Edn1^H/+^* dams at 7.5 dpc. (**A**) Representative images of implantation sites from four groups of dams. H&E staining. Note: Asymmetrical blood vessels in *Edn1^H/+^* dams. (**B**) CD31 (PECAM-1, a marker for endothelial cells) immunostaining (brown) showed decreased density in *Edn1^H/+^* dams compared with WT dams, which was increased by NAM treatment. (**C**) Semi-quantification of CD31^+^ using Image J (Version 1.54p) *n* = 3. Tukey-Kramer HSD. (**D**) Cytokeratin 17 (a marker for trophoblast cells) immunostaining (brown) revealed a blunt and irregular invasion of ectoplacental cone cells in comparison with WT having trophoblast cells sharply invading. NAM corrected this abnormality in *Edn1^H/+^* dams.

**Figure 3 cells-14-01645-f003:**
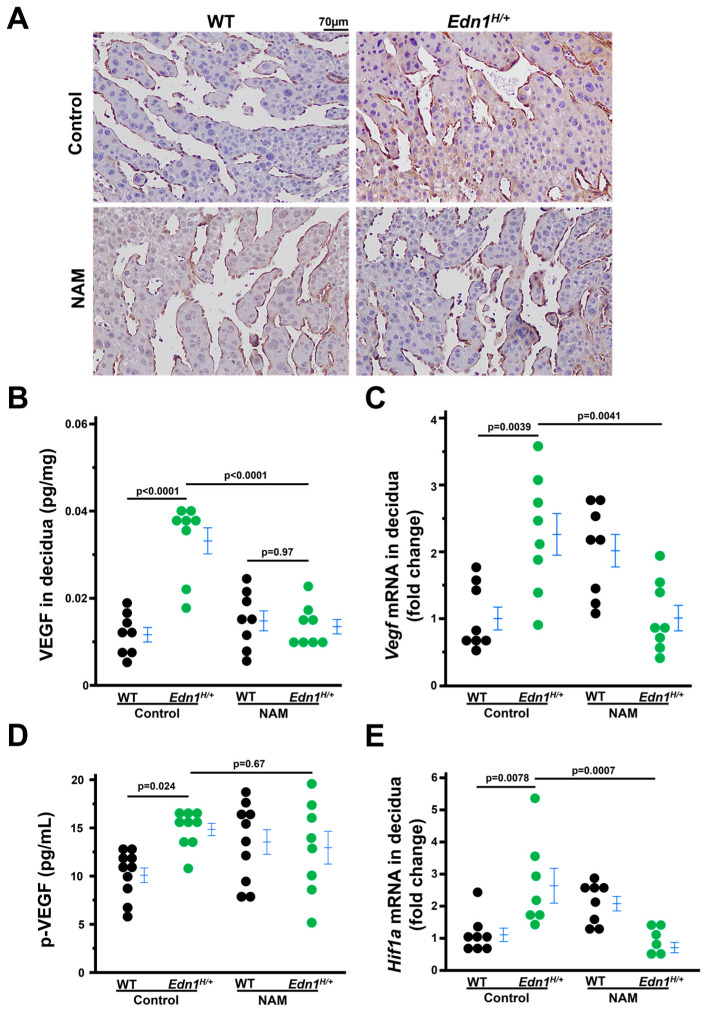
Nicotinamide (NAM) decreases elevated vascular endothelial growth factor A (VEGF) in deciduae from *Edn1^H/+^* dams at 7.5 dpc. (**A**) Representative images of immunostaining of VEGF (brown) in deciduae from four groups of dams. (**B**) VEGF in deciduae from four groups of dams. *n* = 8. Two tail Tukey–Kramer HSD. (**C**) The mRNA levels of *Vegf* in deciduae from four groups of dams. *n* = 8. Tukey–Kramer HSD. (**D**) Plasma (*p*) levels of VEGF from four groups of dams. *n* = 8–10. Tukey–Kramer HSD. (**E**) The mRNA levels of *Hif1a* in deciduae from four groups of dams. *n* = 7 *Edn1^H/+^* groups and *n* = 8 in other three groups. Tukey-Kramer HSD.

**Figure 4 cells-14-01645-f004:**
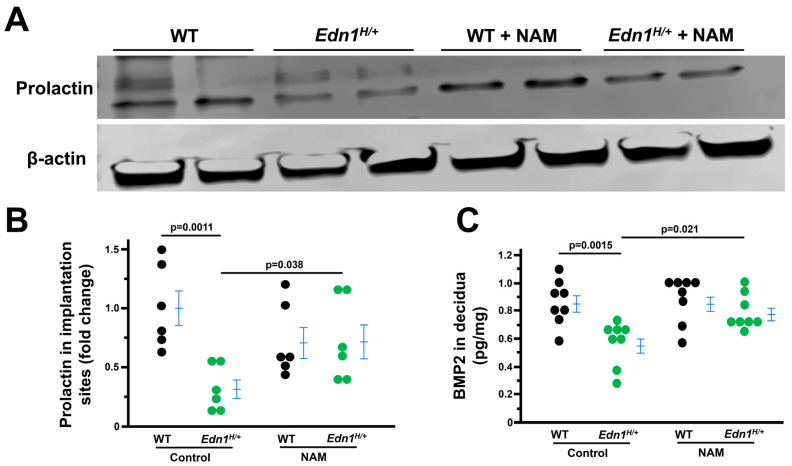
Nicotinamide (NAM) corrects prolactin and BMP2 in the deciduae from *Edn1^H/+^* dams at 7.5 dpc. (**A**) Western blot of prolactin. (**B**) densitometric quantitation of prolactin. *n* = 6. Two-tail Student t-test (**C**). BMP2 in the deciduae from four groups of dams. *n* = 8. Tukey–Kramer HSD. Note: Prolactin and BMP2 (Bone Morphogenetic Protein 2) are markers of decidualization.

**Figure 5 cells-14-01645-f005:**
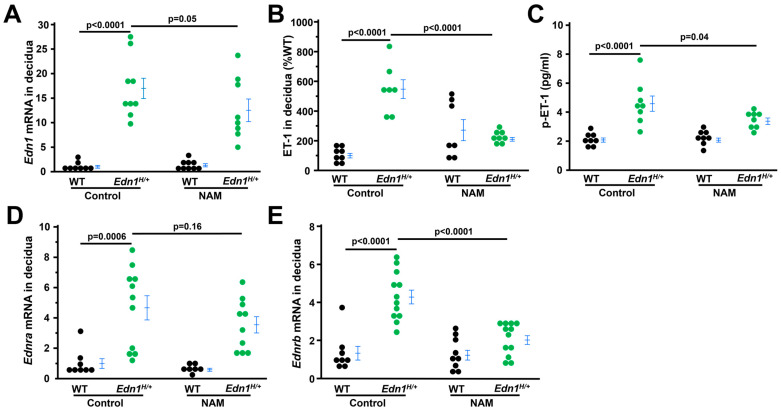
Nicotinamide (NAM) alters the expression of ET-1, EDNRA and EDNRB in the deciduae from *Edn1^H/+^* dams at 7.5 dpc. (**A**) Decidual mRNA levels of *Edn1* in four groups of mice. *n* = 8–9. (**B**) ET-1 levels in the deciduae from four groups of dams. Note: NAM decreased ET-1 more than 50%. *n* = 7–8. (**C**) The concentration of ET-1 in plasma (*p*). *n* = 8. (**D**,**E**) Decidual mRNA levels of *Ednra* and *Ednrb* in four groups of mice. *n* = 8–9. Note: NAM decreased the expression of *Ednrb* more than 50%. *n* = 8–12. Tukey–Kramer HSD.

**Figure 6 cells-14-01645-f006:**
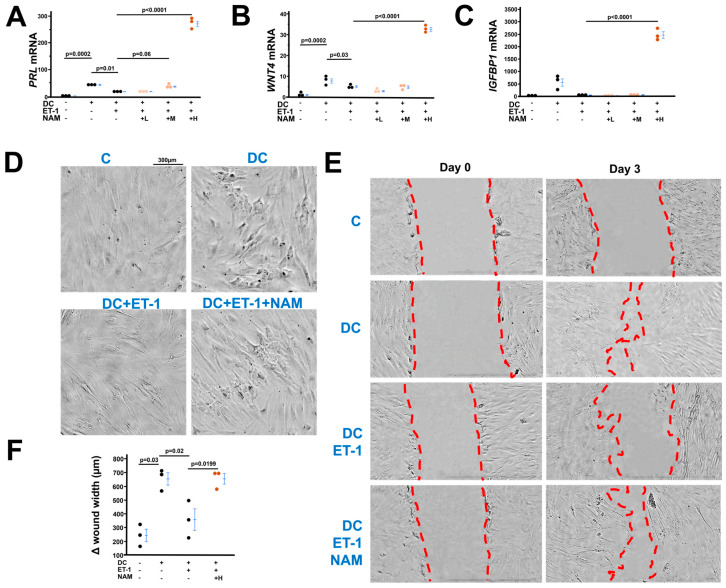
Nicotinamide (NAM) counteracts the inhibitory effects of ET-1 on endometrial stromal cells differentiation. (**A**–**C**) The expression of decidualization markers (PRL, WNT4, IGFBP1) was upregulated three days after a decidualization cocktail (DC) exposure. L: low dose of NAM (0.1 mM), M: medium dose of NAM (1 mM). H: high dose of NAM (10 mM). *n* = 3. Tukey–Kramer HSD. (**D**) The morphology of stromal cells six days after decidualization. Control (**C**) cells had a typically fibroblastic appearance. Cells with DC had characteristics of decidual cells: larger and multinucleated. Cells with DC and ET-1 exhibited more fibroblastic-like features. NAM restored the characteristics of decidual cells in cells with DC and ET-1. (Appendix A showing larger picture). (**E**) images of a wound healing assay in HESCs. Dotted lines indicate location of the scrape. (Nikon TMS microscopy, 4×) (**F**) Δ wound width = wound width at day 0 − wound width at day 0. *n* = 3. Tukey-Kramer HSD.

**Figure 7 cells-14-01645-f007:**
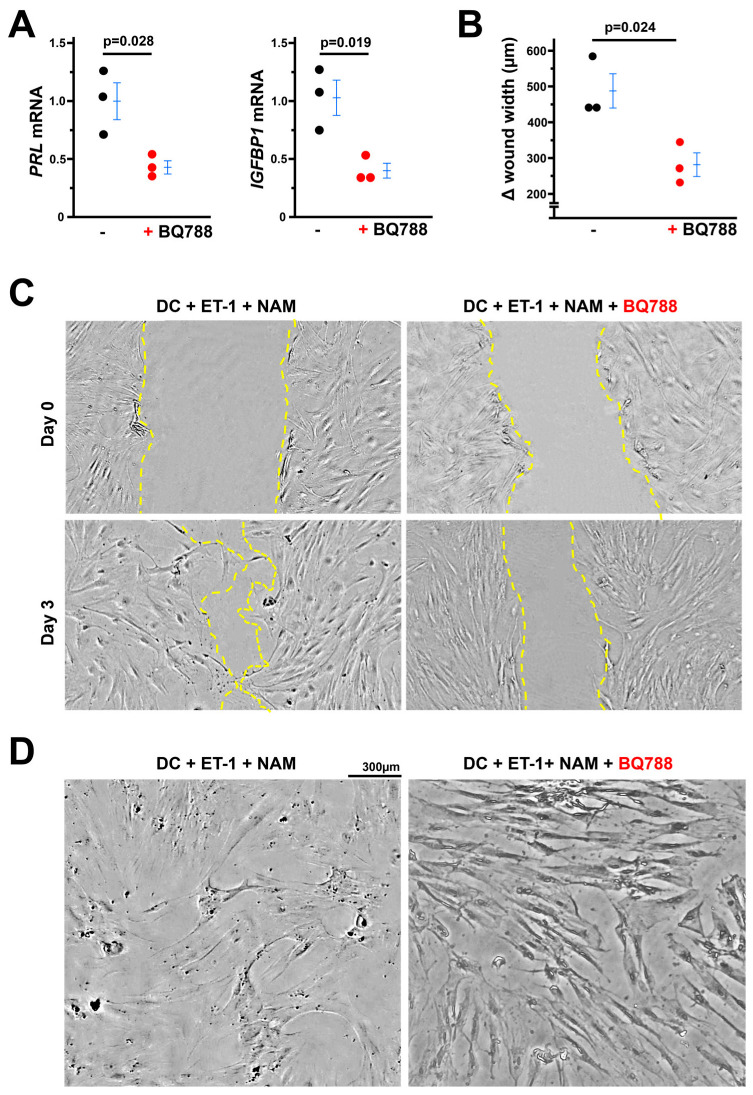
BQ788 (a selective EDNRB antagonist) abolishes the effect of NAM on the stromal cell decidualization. (**A**) BQ788 decreased the expression of markers of decidualization. -: cells exposed to decidual cocktail (DC) + ET-1 + NAM, +: cells exposed to decidual cocktail (DC) + ET-1 + NAM + BQ788. *n* = 3. Two-tailed *t* test. (**B**,**C**) BQ788 inhibited stromal cell motility. Δ wound width = wound width at day 0 − wound width at day 0. *n* = 3. Two-tailed *t* test. (**C**) Representative pictures of stromal cells at different stages of decidualization: day 0 or day 3. (Nikon TMS microscopy, 4×). Yellow dotted lines indicate location of the scrape. (**D**) BQ788 impaired morphological changes six days after decidualization induced by DC.

## Data Availability

RNA sequencing data are deposited to NCBI Gene Expression Omnibus (accession #: GSE306844).

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
