# Peer review of "Nicotinamide Counteracts the Detrimental Effect of Endothelin-1 on Uterine Decidualization During Early Pregnancy by Influencing EDNRB"

_cells, 2025, doi:10.3390/cells14211645_

Round 1
Reviewer 1 Report
Comments and Suggestions for Authors
- Line 2: Please re-word the manuscript title to remove the possessive form of endothelin-1. Please consider an alternative, i.e., Nicotinamide counteracts the detrimental effect of endothelin-1 on uterine decidualization during early pregnancy by influencing EDNRB. Please also amend the possessive form of endothelin-1 in the abstract and throughout the manuscript.
- Abstract, line 16: If endothelin-1 is to be used in its abbreviated form (ET-1) in the abstract, please define it at first mention, i.e., at line 16, not at line 28, and use the abbreviated form thereafter.
- Line 31: Please consider spelling out endothelin-1 as a keyword.
- Line 36: Consider replacing the word "executes" with "exerts" or something similar
- Lines 39–40: Consider describing preeclampsia as "...characterized by hypertension..."
- An overall grammatical check will greatly improve the manuscript (e.g., consider writing in past tense)
- Line 135: May you clarify why 0.01 M of ET-1was deemed the optimal dose? How does this concentration compare to physiological levels in humans?
- Lines 146–153: As HESCs are highly proliferative, how did you account for cellular proliferation into the scratch wound versus cellular migration into the scratch would? Using 2% charcoal-stripped calf serum can impede both cellular proliferation and migration. Example, to measure only cell migration, did you treat the cells with a proliferation inhibitor before or after the scratch? If so, please include this data. If not, please discuss this as a potential limitation of the study.
- Line 169: Please replace "violently" with "vigorously"
- Lines 204–205; 211; 227–228; 249–250; 271–272: Could the authors elaborate (in the discussion section) on why they suppose NAM had no effect on the wild type dams as it relates to implantation site volume/numbers, decidual vascular structure; VEGF expression; prolactin levels, etc.? In other words, please discuss the apparent neutral effect of NAM in your experimental set up.
- By what mechanism of action do you propose that NAM is exerting its beneficial effects for the Edn1H/+ dams? Please add this to the discussion.
Author Response
Thank you for reviewing our manuscript. When the editor reformats the manuscript, the line numbers will change. Please see yellow highlighted parts for our revisions.
1. Line 2: Please re-word the manuscript title to remove the possessive form of endothelin-1. Please consider an alternative, i.e., Nicotinamide counteracts the detrimental effect of endothelin-1 on uterine decidualization during early pregnancy by influencing EDNRB. Please also amend the possessive form of endothelin-1 in the abstract and throughout the manuscript.
Reply: We have fixed the possessive form of endothelin-1 throughout the manuscript.
2. Abstract, line 16: If endothelin-1 is to be used in its abbreviated form (ET-1) in the abstract, please define it at first mention, i.e., at line 16, not at line 28, and use the abbreviated form thereafter.
Reply: We have added the abbreviated form of ET-1 at line 16, and we fixed the rest of the abstract.
3. Line 31: Please consider spelling out endothelin-1 as a keyword.
Reply: We changed it to the full name.
4. Line 36: Consider replacing the word "executes" with "exerts" or something similar
Reply: We changed it to “exerts”.
5. Lines 39–40: Consider describing preeclampsia as "...characterized by hypertension..."
Reply: We deleted the previous redundant description of preeclampsia and adopted your suggestion.
6. An overall grammatical check will greatly improve the manuscript (e.g., consider writing in past tense)
Reply: We checked the overall grammar again and used past tense. Changes were highlighted in yellow.
7. Line 135: May you clarify why 0.01 M of ET-1was deemed the optimal dose? How does this concentration compare to physiological levels in humans?
Reply: We performed RNA-seq on HESC cells treated with three doses of ET-1 (0.01, 0.1, and 1 μM). We found out that all three doses exert similar effects on the expression of decidualization markers. Based on this, we chose the lowest dose possible, which is 0.01 μM. The physiological concentration of ET-1 in preeclamptic patients is 20-50 pg/mL (PMID: 15840756), corresponding to 8-16 pM. ET-1 biological function is thought to act locally in a paracrine/autocrine fashion rather than as an endocrine. Circulating plasma ET-1 level does not appear to be a good indicator of ET-1 activities; ET-1 is rapidly cleared from circulating via its clearance receptor, EDNRB (PMID: 8147891).
8. Lines 146–153: As HESCs are highly proliferative, how did you account for cellular proliferation into the scratch wound versus cellular migration into the scratch would? Using 2% charcoal-stripped calf serum can impede both cellular proliferation and migration. Example, to measure only cell migration, did you treat the cells with a proliferation inhibitor before or after the scratch? If so, please include this data. If not, please discuss this as a potential limitation of the study.
Reply: We maintained the HESC cells using 10% charcoal-stripped fetal bovine serum and their doubling time is roughly 40 hours. During the wound-healing assay, we switched to 2% charcoal-stripped fetal bovine serum to reduce proliferation. Furthermore, our RNA-seq data showed that decidualization cocktail significantly reduces the proliferative ability of HESC cells by downregulating Id2 and Mki67. Our wound-healing results are consistent with the report by Yamaguchi et al (PMID: 40085209). We have made a figure using the RNA-seq data. The figure is attached below.
9. Line 169: Please replace "violently" with "vigorously"
Reply: We replaced it with “vigorously”.
10. Lines 204–205; 211; 227–228; 249–250; 271–272: Could the authors elaborate (in the discussion section) on why they suppose NAM had no effect on the wild type dams 4 relates to implantation site volume/numbers, decidual vascular structure; VEGF expression; prolactin levels, etc.? In other words, please discuss the apparent neutral effect of NAM in your experimental set up.
Reply: In the current study, NAM had no effect on WT dams which was consistent with our prior reports (PMID: 27821757, 32905409) and others (PMID: 27927652, 33440677). The known mechanism by which NAM exerts its beneficial effects is suppressing mobilization of Ca2+ from sarcoplasmic reticulum into cytoplasm through inhibiting ADP-ribose cyclase activated by ET-1. The potential mechanism we proposed in this study is that NAM corrects EDNRB expression. In WT dams, ET-1 and its downstream ADP-ribose cyclase and EDNRB expression are in baseline levels (not activated), so there is no elevated mechanism for NAM to counteract. This is a possible reason that there is no obvious effect of NAM on WT dams, as there is no underlying issue for NAM to correct.
11. By what mechanism of action do you propose that NAM is exerting its beneficial effects for the Edn1H/+ dams? Please add this to the discussion.
Reply: The overall effect of NAM is beneficial and possibly through modulating EDNRB-related signaling pathway. Comprehensive stereological and/or single cell RNA-seq assessment of the effects of ET-1/NAM on different compartments of the deciduae, will elucidate how ET-1/NAM influence each type of cells and their crosstalk, which will provide deeper insight into the mechanisms underlying NAM’s beneficial effects.
Reviewer 2 Report
Comments and Suggestions for Authors
Manuscript ID: cells-3926256
In the current manuscript Wang et al. assessed the impact of endothelin-1 (ET-1) on decidualization, and the effect of nicotinamide (inhibitor of endothelin-1). Authors used animal model of preeclampsia-like symptoms (wild type (WT) and Edn1H/+ female mice) and demonstrated that Edn1H/+ dams have smaller implantation site volume that as ameliorated with nicotinamide treatment. One of the strengths of the current study is the application of animal model of pregnancy complication together with in vitro assessment. The manuscript is logically organized, and methods are used as appropriate. There are several minor comments that can be addressed in order to improve manuscript quality (presented below):
Comments to authors:
- Please include information on animal age in methods section;
- Western blotting: please include information on dilution of primary/secondary antibody, time of incubation and conditions of incubation;
- Can authors speculate in the discussion section on what might be a potential role of EDNRB in decidualization? Including the information on what types of cells decidua composed of and its relative expression on those cells in healthy pregnancy vs preeclampsia. Are their function might be different depending on cellular type it is expressed on?
- Line 412-413: please provide reference for statement.
Author Response
Thank you for reviewing our manuscript. When the editor reformats the manuscript, the line numbers will change. Please see green highlighted parts for our revisions.
1. Please include information on animal age in methods section;
Reply: We have added animal age. Please see the yellow highlighted part in section 2.2.
2. Western blotting: please include information on dilution of primary/secondary antibody, time of incubation and conditions of incubation;
Reply: We have added the relevant details in the method section. Please see the yellow highlighted part in section 2.7.
3. Can authors speculate in the discussion section on what might be a potential role of EDNRB in decidualization? Including the information on what types of cells decidua composed of and its relative expression on those cells in healthy pregnancy vs preeclampsia. Are their function might be different depending on cellular type it is expressed on?
Reply: scRNA-seq analysis of implantation site of WT mouse uterus at the invasion phase of embryo implantation approximately 7.5 dpc revealed multiple types of cells including stromal cell, epithelial cells, endothelial cells, immune cells (PMID: 35123575). All these types of cells express EDNRB, however, the relative expression of EDNRB in normal or preeclamptic pregnancy is not clear. The in vitro data we obtained here suggests that EDNRB expression is upregulated during normal stromal cell decidualization and this upregulation is inhibited by ET-1 exposure. In contrast, the in vivo data showed Edn1H/+ dams had increased decidual EDNRB expression. Taken together, these data suggest EDNRB signaling plays distinct role in different types of cells in normal pregnancy or PE, especially in a subset of PE associated with abnormal high ET-1.
4. Line 412-413: please provide reference for statement.
Reply: To the best of our knowledge, we are the first to report that NAM influences EDNRB in the current manuscript. Therefore, there is no reference for us to cite given the limited information currently available.
Round 2
Reviewer 1 Report
Comments and Suggestions for Authors
Thank you for your responses. After Editorial review and updates, I am happy to see this published.